# Brain Metastasis: A Literary Review of the Possible Relationship Between Hypoxia and Angiogenesis in the Growth of Metastatic Brain Tumors

**DOI:** 10.3390/ijms26157541

**Published:** 2025-08-05

**Authors:** Lara Colby, Caroline Preskitt, Jennifer S. Ho, Karl Balsara, Dee Wu

**Affiliations:** 1Department of Radiological Sciences, University of Oklahoma Health Sciences, Oklahoma City, OK 73104, USA; lara-colby@ouhsc.edu (L.C.); carlie.preskitt@gmail.com (C.P.); jennifer-s-ho@ouhsc.edu (J.S.H.); 2Department of Neurosurgery, University of Oklahoma Health Sciences, Oklahoma City, OK 73104, USA; 3Department of Computer Sciences, University of Oklahoma, Norman, OK 73019, USA

**Keywords:** brain metastasis, hypoxia, angiogenesis, VEGF, HIF1α

## Abstract

Brain metastases are a common and deadly complication of many primary tumors. The progression of these tumors is poorly understood, and treatment options are limited. Two important components of tumor growth are hypoxia and angiogenesis. We conducted a review to look at the possibility of a symbiotic relationship between two transcription factors, Hypoxia-Inducible Factor 1α (HIF1α) and Vascular Endothelial Growth Factor (VEGF), and the role they play in metastasis to the brain. We delve further into this possible relationship by examining commonly used chemotherapeutic agents and their targets. Through an extensive literature review, we identified articles that provided evidence of a strong connection between these transcription factors and the growth of brain metastases, many highlighting a symbiotic relationship. Further supporting this, combinations of chemotherapeutic drugs with varying targets have increased the efficacy of treatment. Angiogenesis and hypoxia have long been known to play a large role in the invasion, growth, and poor outcomes of tumors. However, it is not fully understood how these factors influence one another during metastases. While prior studies have investigated the effects separately, we specifically delve into the synergistic and compounding effects that may exist between them. Our findings underscore the need for greater research allocation to investigate the possible symbiotic relationship between angiogenesis and hypoxia in brain metastasis.

## 1. Introduction

Brain metastases are the most common form of brain tumors in adults, with the most prevalent origins being the lung, breast, skin, and colon [1,2]. In a 2024 review, Gomez explains that a significant proportion of brain metastasis cases originate from primary lung and breast cancers, which account for 40% and 20% of cases, respectively. Lung and bronchus cancers have the highest incidence of brain metastases, with 7.1 cases per 100,000 persons reported in one study [1]. Treatment options for brain metastasis remain limited, with local treatments, such as stereotactic radiotherapy or combination chemotherapy regimens, being the mainstays of treatment. Although useful, these treatment modalities show minimal improvement, poorer outcomes, and higher mortality rates than when used on their primary tumor counterparts [3,4]. Due to this, brain metastases have been receiving increasing levels of attention, with more research dedicated to treatment options in recent years [1]. Despite this greater allocation of time and resources, there remain gaps in understanding how brain metastases grow and proliferate once they have taken occupancy within the brain.

There are four mandatory, chronological steps for metastasis to proceed successfully: seeding of the blood vessels, early passage into the surrounding tissue, rapid perivascular growth, and early co-option and angiogenesis (Figure 1) [2,5]. Seeding of blood vessels in metastases refers to the migration of cancer cells from the primary tumor to distant organs, where they can establish new growths. This process involves cancer cells entering the bloodstream, traveling to other parts of the body, and exiting the blood vessels to form new tumors. Vascular co-option in brain metastases refers to the process by which cancer cells utilize existing blood vessels to grow and spread throughout the body. Unlike angiogenesis, which involves the creation of new blood vessels, vascular co-option involves cancer cells interacting with pre-existing vessels. This interaction helps cancer cells evade the immune system and may be a critical step in the early stages of brain metastasis, allowing them to establish themselves in the brain tissue after crossing the blood–brain barrier.

Once the tumor has taken root, growth proceeds at a rapid pace. This is first accomplished via theft of the nutrients and oxygen from the nearby vasculature, otherwise known as co-option [3,6]. These co-opted vessels undergo certain changes, becoming tortuous, elongated, and dilated, ultimately leading to a reduction in available areas for oxygen and nutrient exchange. This rearrangement and utilization of the surrounding blood supply allows for brain metastases to quickly adhere, proliferate, and grow within their new environment, as well as maintain and deliver oxygen and vital nutrients to the growing tumor [6]. A study completed by Carbonell et al. used in vivo experimental models of mice to demonstrate that tumor cell growth was able to be observed without evidence of any new vasculature, providing evidence that co-option had taken place [6]. However, there is a maximum growth point, as this simple diffusion can only occur up to a maximum distance of two millimeters [7]. Once tumor growth exceeds this point, co-option can no longer provide enough oxygen or nutrients, and hypoxia develops. This lack of oxygen will quickly lead to regression of the tumor if not quickly corrected or circumvented [8]. Brain metastases are able to circumvent this setback by switching from co-option to angiogenesis, allowing for continued growth past the confines of the new environment [2,8]. Angiogenesis occurs via a new vascular architecture that grows from within the tumor or just adjacent to the tumor. This process is known as the angiogenic switch (Figure 2).

While normal healthy cells stop proliferating under hypoxic conditions, tumor cells violate this rule by continuing to proliferate, with some areas being extremely well perfused, allowing for quicker growth [9,10]. This rapid growth is in part allowed for by Hypoxia Inducible Factor 1α (HIF1α), a transcription factor that is normally rapidly degraded by proteasomes in an oxygen-rich environment. However, under hypoxic conditions, HIF1α becomes stabilized and is transported to the nucleus to interact with the HIF response elements that modulate transcription [11]. Another major role of HIF1α is the upregulation of the Vascular Endothelial Growth Factor Receptor (VEGFR), ultimately leading to the overexpression of VEGF in tumor cells [8,10,12]. One important subtype, VEGF-A, attracts and guides sprouting neovessels into areas that are low in oxygen, supporting the possibility of a symbiotic or otherwise circular relationship between the two transcription factors [8]. According to Lee et al., oxygenases and oxidases can play roles in cellular responses to hypoxia [13]. However, the activity of these oxygen-dependent enzymes can be reduced under hypoxia, leading to various cellular consequences, which can include changes in metabolism, protein stability, and folding. Acute cellular responses to hypoxia inhibit mitochondrial activity, reducing ATP production and increasing reactive oxygen species (ROS) production. Long-term cellular responses can include changes through transcriptional regulation via HIFs and UPR-related transcription factors (ATF4, XBP1, ATF6). These changes promote angiogenesis, as well as chromatin remodeling, according to Lee. Thus, HIF and VEGF coexist in hypoxic tumor environments, driving processes such as angiogenesis, migration, invasion, and immune evasion [14]. Despite HIF1α being incredibly important in the growth of brain metastasis, its overall role in hypoxic signaling is not as well understood [14].

There are several chemotherapeutic drugs used in the treatment of glioblastomas, hemangioblastomas, gliomas, and, to a lesser extent, brain metastases. These therapeutic agents include Temozolomide (TMZ), an alkylating agent that induces DNA damage and causes cell cycle arrest [15,16]; Bevacizumab, a monoclonal antibody against VEGF [17]; Topotecan and Irinotecan, selective inhibitors of Topoisomerase 1 and HIF1α [18]; and Belzutifan, a selective inhibitor of HIF2α used to treat VHL-associated hemangioblastomas [19]. Because each agent targets a different component of brain tumors, they are shown to have increased effects when combined with each other or with other drugs affecting angiogenesis or hypoxia levels [15,16,17,18,20]. This increased efficacy highlights the heterogeneity in tumor cells, as different components of the tumor often utilize both hypoxic conditions and angiogenesis in varying ratios [7,21,22,23].

This paper provides a review of the existing literature that supports the possibility of brain metastases establishing a symbiotic relationship between hypoxia and angiogenesis. We aim to further define the relationship that is characterized by the tumor intentionally maintaining a hypoxic environment to allow for continuous stimulation of angiogenesis via HIF1α and VEGF. We investigate this relationship by examining studies of commonly used chemotherapeutic protocols on various brain tumors and how they target both transcription factors as a way to provide evidence for this relationship.

## 2. Results

### 2.1. HIF1α and Angiogenesis

There were eighteen studies specifically related to angiogenesis, hypoxia, and brain metastasis that discussed the relationships between HIF1α and VEGF, as noted in Table 1. Spanberger et al. performed a retrospective analysis of the effect of edema on angiogenesis and HIF1α, which hypothesized that larger amounts of edema were due to increased angiogenesis and vasculature, causing vessels to become more prone to leaks. Patients with less edema demonstrated lower HIF1α expression and less angiogenic activity [23]. Liu et al. examined anti-inflammatory microglia activated by HIF1α, which allowed for increased growth and, consequently, a poorer prognosis. Liu also investigated radiation in NSCLC, leading to a downregulation of HIF1α seven and fourteen days post-radiation [4]. Bergoff et al. measured Ki67 (a marker of proliferation), HIF1α, and CD34 (a measure of hematopoietic stem cells) of Non-Small Cell Lung Cancer (NSCLC). There was a weak correlation found between proliferation and HIF1α, with lower levels of Ki67 and HIF1α leading to better overall survival [24]. Ebright et al. investigated RNA sequencing on breast cancer strains, finding that there were increasing rates of growth with increasing levels of HIF1α. HIF1α was also found to be increased in brain metastases compared to primary tumors [14]. Anuja et al. investigated the role of HIF reprogramming of cellular metabolism in the angiogenesis and energy metabolism of solid tumors. Their results illustrated that these changes allowed for the rapid adaptation and increased resistance to a variety of treatment modalities. All of these effects ultimately led to increased vascular permeability, alteration of the extracellular membrane, and increased immunosuppression [10]. The correlation and findings in each of these studies suggest that HIF1α expression has a direct effect on the activation of angiogenesis and proliferation within the tumor environment and that the expression of a high amount of HIF1α is necessary to support continued tumor growth and survival.

### 2.2. HIF1α and Hypoxia

Heddleston et al. evaluated cellular responses to oxygen by monitoring HIF1α and its transcriptional activity. They hypothesized that with the increasing level of HIF1α altering the microenvironment of the tumor and surrounding area, the epigenetic makeup of the tumor was able to mutate. Heddleston found that increased HIF1α leads to an increased level of cancer stem cells (CSCs), which allows for the rapid growth of tumors and increased levels of VEGF. Thus, this supports our theory that under hypoxic conditions, tumors undergo an angiogenic switch to support their continued survival [22]. Shin et al. studied the amount of HIF1α and HIF2α in tumor cells by measuring their protein and mRNA levels. Both factors are readily destroyed via proteasomes under normoxic conditions but become stabilized when oxygen becomes diminished. They found an increase in both HIF1α and HIF2α in a wide variety of brain metastases [11]. Ruan et al. explained the role of HIF1α in human cancer and how an increasing tumor size leads to a larger internal hypoxic environment as well as a corresponding increase in vascular density. This growth is allowed to happen because the tumor is able to adapt to the absence of growth signals normally present, as well as becoming resistant to antiproliferative signals. This overall process leads to an upregulation of proangiogenic factors as well as a downregulation of antiangiogenic factors [8]. Lee et al. investigated biopsy samples of brain metastases from Small Cell Lung Cancer (SCLC) to determine the influence of HIF1α on tumor growth. The samples showed the influence of HIF1α on tumor initiation, growth, invasion, and metastasis. However, they did not find an association between the expression of HIF1α and clinical outcomes [13]. da Ponte et al. evaluated the interrelation between hypoxia and angiogenesis, finding a strong correlation between the factors in their cohort of 23 glioblastoma (GBM) patients. Their study also provided insight into the heterogeneous makeup of tumors, using fluorescent imaging modalities to note areas of varying perfusion within tumors and between patients [9]. Zhong et al. screened different cancer levels of HIF1α by utilizing immunohistochemistry, finding that in primary tumors that have metastasized, there is an overabundance of HIF1α. They also found that in different tumors, such as hemangioblastomas, hypoxia could not be attributed to the distance from a blood vessel or lack of available nutrients or oxygen, emphasizing the idea that tumors intentionally maintain a heterogeneous environment of both hypoxia and angiogenesis [12].

### 2.3. VEGF and Hypoxia

Kim et al. investigated the pathogenesis of brain metastasis in breast cancer by injecting metastatic and non-metastatic cells into the carotid arteries of nude mice and then measuring the mean survival, with the mice with brain metastases having a significantly shorter survival. Kim illustrated that metastatic cells had elevated levels of VEGF-A when compared to primary tumors, even when VEGF was cultured in normoxic conditions. The levels of angiogenesis were also significantly decreased when VEGF inhibitors were administered to the mice with brain metastasis. Kim also found no difference in HIF1α levels between primary and metastatic tumors from breast cancer [25]. Ban et al. investigated three different VEGF inhibitors, namely, AAL993, SU5416, and KRN633, and their ability to inhibit not only VEGFR but also HIF1α. AAL993 was observed to have the ability to lower HIF1α enough to also inhibit VEGF, and all three inhibitors were able to suppress HIF1α under hypoxic conditions. This illustrates the close relationship between VEGF and HIF1α, with one inhibition leading to the downregulation or inhibition of the other [26]. Crowder et al. investigated CSCs in brain metastasis from breast cancer under hypoxic conditions. Their findings showed that hypoxic stabilization of HIF1α led to stimulation of CSCs, allowing for metastatic transformation, as well as increased survival and decreased effectiveness of treatments [27].

### 2.4. Other Factors in Hypoxia

Considering a broader spectrum of factors that have a direct influence on hypoxia, Schiefer et al. investigated the importance of Lactate Dehydrogenase A (LDHA). This enzyme catalyzes the conversion from pyruvate to lactate in brain metastasis with primary tumors originating from melanomas. LDHA levels were found to correspond to hypoxia levels, with more necrotic tumors and increased distance from vasculature having increased levels of the enzyme [30]. Delaney et al. investigated the radiation resistance of brain metastases with primary tumors. They injected oxygen microbubbles into nude mice who had been inoculated with breast cancer. Within the parameters of this experiment, there was an improvement in the survival rate of the mice that had been given oxygen before undergoing radiation therapy, suggesting that the presence of oxygen prevented the cancer from continuing to survive amidst radiation. However, a drawback of their study is the modest size of the control group [31]. This suggests that tumors may, in fact, thrive in the absence of oxygen because of their ability to trigger an angiogenic cascade to obtain necessary nutrients via HIF1α and other factors.

Corroyer et al. performed brain biopsies on 28 patients with brain metastasis from primary tumors originating in the lung. They observed the carbonic anhydrase-IX (CA-IX) level, which is a marker of HIF1α. Their findings showed increased levels of CA-IX in 22 of the 28 tumors, with an increase in tumor heterogeneity also observed [20]. Additionally, their study provided insight into location-dependent hypoxia, as cortical metastases exhibited less vascularization and more hypoxia staining compared to striatal metastases. Location-dependent hypoxia heterogeneity was suggested to have implications for radiotherapy efficacy, which is a unique insight in considering therapeutic protocols for better patient outcomes. In their study, standard whole-brain RT was less effective in controlling hypoxic cortical metastases, leading to recurrence specifically in these tumors [20]. Sonveaux et al. investigated lactate produced by hypoxic regions of tumors and how this fuels oxidative metabolism, creating a “symbiotic effect” [21]. The elevation of lactate led to increased metastasis growth and lower survival of the host. The lactate was then shown to be shuttled via channels to oxygenated portions of tumors, highlighting a reliance on oxidative metabolism, even in oxygen-rich areas of brain mets. Overall, this allows for a bypass of the tumor by relying on specific amounts of oxygen [21]. These studies not only provide evidence that tumors are able to circumvent environmental limitations by continuing to grow despite a lack of oxygen, but they also provide evidence that maintaining a hypoxic environment is overall beneficial to the tumor as it triggers angiogenesis.

### 2.5. Therapeutic Protocols

There were six articles in our review that discussed common chemotherapy medications and their effects on hypoxia and angiogenesis. Tuettenberg et al. investigated how continuous low doses of the chemotherapeutic drug Temozolomide (TMZ) in combination with the cyclooxygenase-2 (COX-2) inhibitor, Rofecoxib, affected angiogenesis in glioblastomas (GBMs), which suggests a possible novel antiangiogenic strategy for treatment [16]. TMZ’s mechanism of action involves alkylating guanine residues, inducing DNA damage, and cell cycle arrest. By inhibiting COX-2, an enzyme that promotes angiogenesis by increasing the expression of proangiogenic growth factors, like VEGF, in addition to inducing cell cycle arrest and DNA damage, the tumor environment can be targeted through two separate mechanisms. Well regarded as one of the most vascularized tumors found in humans, a GBM is characterized by its extensive angiogenesis, one of the most significant hallmarks found that is crucial to its development and progression [32]. Chemotherapeutic drugs wield a cytotoxic effect on tumor cells along with any antiangiogenic activity through interference with endothelial cell proliferation. However, the effects remain slim due to a traditional cyclic high-dose scheduling, where endothelial cells gain enough time to repair the damage induced by chemotherapy. Birthed from these observations, the concept of “antiangiogenic scheduling” was developed, where cytotoxic chemotherapeutic drugs are given continuously at low doses to maximize the effects on tumor endothelial cells [33].

In addition, the anti-inflammatory action of the COX-2-specific inhibitors Rofecoxib or Celecoxib has been shown to induce an antiangiogenic effect in vitro and in vivo [29]. Two forms exist, with COX-1 being constitutively expressed in a range of tissues; meanwhile, COX-2 is strongly expressed in human tumors, correlates with tumor angiogenic activity, and is cytokine-inducible [16]. Most notably, the antiangiogenic activity of COX-2 inhibitors stems from the downregulation of the crucial angiogenic growth factor, VEGF, which thwarts endothelial cell proliferation and ushers in endothelial cell apoptosis. Because of the culmination of combined effects, COX-2 inhibitors were selected as adjuvant compounds to augment the efficacy of continuous low-dose chemotherapy. Through histoanalysis of tumor specimens demonstrating antiangiogenic efficacy of continuous low-dose TMZ and Rofecoxib in GBM patients, Tuettenberg et al. substantiated that patients with higher vessel density were correlated with significantly better tumor control than patients with lower vessel density. This indicated that the response to treatment depended on the angiogenic activity of the individual GBM. Thus, patients with highly angiogenic tumors would be the most suitable candidates [16]. Ultimately, it was shown that the antiangiogenic effect of the two medications combined decreased the level of density in highly angiogenic tumors. A limitation of their study was a small patient sample and studies with small tumor loads, such as after partial resection of GBMs [16].

Brain tumors are known to be the most vascularized solid tumors found in humans, with angiogenesis playing a key role in driving tumor progression. Peleli et al. investigated the mechanisms surrounding excessive vascularization of malignancies in the brain. Thus far, antiangiogenic therapies have been trialed with limited or no significant improvement in regard to overall survival. Since 2009, only Bevacizumab (Bev), a human monoclonal antibody that negates VEGF-A activity and harbors antiangiogenic effects, has been approved by the FDA. Though Bev substantially improves progression-free survival for 6 months, it does not improve overall survival. The most compelling explanation for these results is that VEGF is one of many growth factors regulating angiogenesis in brain tumors [15]. High levels of hypoxia characterize most brain and CNS tumors, leading to reduced efficacy of Bev. The molecular mechanism involves hypoxia mediating the upregulation of the HIG2 gene or the downregulation of the CYLD gene. The gene HIG2 encodes for a protein that induces increased HIF-1β, VEGF expression, and Bev resistance. Meanwhile, hypoxia suppresses the gene CYLD, leading to significant inflammation and reduced long-term efficacy of Bev. Peleli et al. emphasized the importance of administering complementary substances that are either hypoxia-resistant or hypoxia-activated, which can then exert a cytotoxic effect. One example of this is TH-302, which is activated under low oxygen and harbors a cytotoxic effect. Alternatively, drugs that target crucial molecular mediators of hypoxia (HIF), such as Amphotericin-B and 2-methoxyestradiol, which possess HIF-inhibitory activity, can also be considered [15]. To overcome the shortfalls of individual drug regimens, future drug strategies need to target more than just one aspect of the tumor [15].

Zhao et al. studied a novel drug combination consisting of Bev and a VEGF-trap in primary tumors and brain metastasis originating from the lung. Their results revealed that the single injection efficacy of AAV2-VEGF-Trap significantly inhibited the growth of the glioma, demonstrating antiangiogenic properties comparable to Bev, the standard anti-VEGF therapy. When administered in combination with TMZ, marked tumor growth inhibition was observed. The combination treatment of AAV2-VEGF-Trap with TMZ led to increased apoptotic tumor cells and reduced microvessel density, highlighting a synergistic antitumor effect. Lastly, sustained expression was shown as the AAV2 vector facilitated prolonged expression of VEGF-Trap in vivo. This resulted in the maintenance of antiangiogenic effects over time with a single injection [17]. Guan et al. investigated the novel theory of combining drugs, such as Topotecan, to decrease the hypoxic environment, leading to an increase in the viability of antiangiogenic drugs, such as Bev. Topotecan, a selective inhibitor of topoisomerase I, disrupts the replication and transcription processes in tumor cells, which leads to cell death. At the same time, it inhibits HIF-1α by affecting RNA transcription [18]. Of note, there remains little information on the mechanisms behind how different antitumor drugs interact [18]. Oronsky et al. studied the anti-VEGF drug RRx-001 to assist in normalizing the vasculature and then added Irinotecan or TMZ. The results demonstrated increased uptake of these two chemotherapies after the addition of the anti-VEGF medication [28].

## 3. Discussion

Brain metastasis has been shown to be incredibly dependent on angiogenesis. As discussed previously, without the generation of new vasculature, brain metastases cannot grow beyond a maximum limit, which is determined by oxygen and nutrient availability. Both factors are imperative to the survival of the tumor. However, these confines can be altered with the induction and stabilization of HIF1α and the initiation of angiogenesis, leading to tumor growth well beyond their original limitations. Many prior studies discussed angiogenesis concerning HIF1α. We found only a few articles that did not include both factors, suggesting a strong association between the two. This relationship is highlighted by the increased inhibition of VEGF, ultimately leading to a decrease in HIF1α, increased proliferation despite an increasing hypoxic environment, and VEGF stabilization of HIF1α [26].

A major component of the hypoxic environment created by brain metastases is the stabilization of HIF1α. In normoxic conditions, the transcription factor HIF1α readily breaks down. However, in hypoxic environments, it becomes stabilized, preventing its breakdown. Once stable, it forms a heterodimeric transcription factor, helping to modulate many aspects of tumor growth [27]. The expression of HIF1α allows for the induction of genes that regulate glucose metabolism, cell growth, apoptosis, angiogenesis, extracellular matrix remodeling, and metastasis [11]. In addition to helping tumor growth, the induction of multiple factors facilitates increased resistance to treatment options, such as chemotherapy and radiation [12,22,31,34], thereby emphasizing the importance of understanding the role of HIF1α in brain metastasis. Previous studies discussed additional key factors that led to the stabilization and increased expression of HIF1α in primary and metastatic tumors, such as midkine [11] or LDHA [30]. Both are heavily influenced by hypoxia. However, it has been shown that in certain brain tumors, distance from the vasculature or varied levels of hypoxia was not always the sole mediator of increased levels of HIF1α [12]. This could delineate the importance of HIF1α in the maintenance of brain metastasis, such that even when not hypoxic from the lack of nearby vasculature, the tumor may favor the factors normally activated by a lack of oxygen and nutrients.

Many prior studies emphasized the heterogeneity in tumors themselves. Concerning angiogenesis and co-option, varying levels of vasculature were found throughout tumors, allowing for increased levels of growth, hypoxia, and induction of HIF1α [9]. Increased HIF1α levels varied between different brain metastases. However, when this transcription factor was expressed, there was an increase in the level of variation within tumors [20]. The findings highlight that even within a similar environment, there is diversity within the tumor itself.

Our findings point to a relationship in which the tumor environment is reliant on multiple transcription factors to promote survival and proliferation. This connection is also highlighted when investigating different chemotherapeutic protocols for various brain tumors. When considering these trials on primary brain tumors, it was shown that treatments became more effective when agents that worked on multiple aspects of the tumors were combined, such as using antiangiogenic RRx-001 before treating with TMZ or Irinotecan, or similarly using Bev before treating with TMZ. These combined protocols lead to decreased angiogenesis and hypoxia and increased uptake of the chemotherapy medications [17,28]. However, the results of this review were mostly focused on primary brain tumors and not specifically on brain metastases. But even within these primary tumors, it was illustrated that a combination of multiple treatments leads to better patient outcomes, as opposed to using a singular chemotherapeutic agent that individually targets angiogenesis, HIF1α, or HIF2α [15,16,17,18,28].

Not every article we examined supported our theory. Some studies found no correlation between the level of HIF1α and the angiogenesis of brain metastasis, while others found opposite effects. Specifically, some studies supported an inverse correlation showing that an increase in HIF1α resulted in decreased angiogenesis [23] or a decrease in the overall prognostic outcome of patients with brain metastasis, with lower amounts of edema and lower amounts of HIF1α [25]. Other studies could not find a correlation between the level of HIF1α and patients’ overall prognoses [13]. However, the level of heterogeneity shown among brain metastases of different patients emphasized the need for more research within this area.

## 4. Materials and Methods

An initial literature search was completed for brain metastasis, angiogenesis, and hypoxia through the Ovid database. The search terms utilized included “brain neoplasms” OR “neoplasm metastasis” AND “cell hypoxia” OR “anoxia” OR “hif” OR “angiogenesis”. The search results were screened using inclusion criteria composed of English articles describing brain metastasis, angiogenesis, or hypoxia. Titles, abstracts, and full articles were screened to include all studies that met the above criteria. Studies that did not describe brain metastasis, angiogenesis, or hypoxia were excluded. Published literature bibliographies were also reviewed to include studies not captured in the initial search. The initial Ovid search included 213 articles, which were screened, and 201 articles were excluded. The bibliography review included 8 articles not found through Ovid. A total of 20 articles were initially reviewed. Upon peer review by the journal, we identified 4 additional articles using the same criteria as the initial search to further support our findings, which have been incorporated into the Results Section.

## 5. Conclusions

Angiogenesis and hypoxia play a joint role in tumor invasion, growth, and overall poor prognostic outcomes for patients whose primary tumors metastasize to the brain. As we found in our study, there is a wide array of research on both HIF1α and VEGF in relation to brain metastasis, as well as research on chemotherapeutic agents and their effects on primary tumors. More promising outcomes have been shown by combining therapeutic agents that target more than one transcription factor, such as anti-VEGF agents combined with hypoxia-reducing agents. However, there remain knowledge gaps in which more research is needed to investigate the relationship between angiogenesis and hypoxia. Does hypoxia followed by vascular growth occur in a stepwise process, or does this heterogeneous occurrence in tumor environments suggest a more symbiotic relationship? We elaborated on this possible interdependent relationship, where the heterogeneous nature of brain metastases is utilized to create a continuous loop, feeding off of the other environments within the tumor to ensure that it continuously expands well past the confines of a healthy cell.

## Figures and Tables

**Figure 1 ijms-26-07541-f001:**
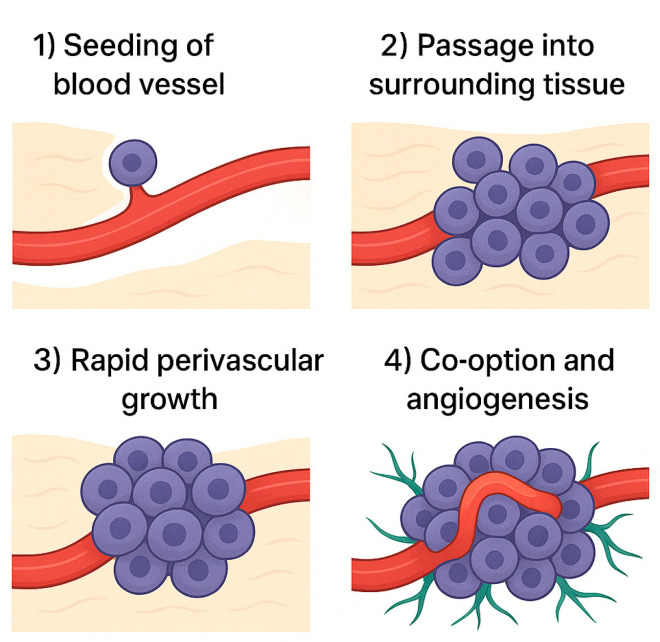
Four steps of tumor metastasis. (**1**) Tumor metastasis takes root in a blood vessel. (**2**) Tumor invades the tissue surrounding a vessel. (**3**) The tumor grows rapidly. (**4**) Co-opted vessels change in size and shape and begin forming new vasculature via angiogenesis.

**Figure 2 ijms-26-07541-f002:**
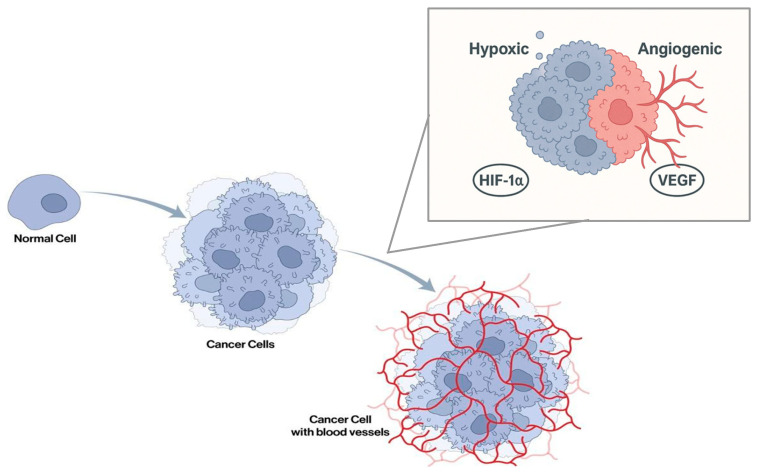
When tumor cells grow past their original confines, they become hypoxic and trigger the activation of HIF1α, leading to the angiogenic switch that allows for the continued growth of blood vessels and the delivery of oxygen and nutrients to the tumor.

**Table 1 ijms-26-07541-t001:** A summary of the total number of articles reviewed in the literature search, the search criteria, and their relation pertaining to our review. This number includes the initial 20 articles as well as the 4 additional articles found in the journal peer review process.

Grouping	Relation	Reference No.
Hypoxia and HIF1α Signaling	Includes sources that focus on the biological effects and molecular mechanisms related to low-oxygen conditions (hypoxia) in tumors and the central transcription factor, Hypoxia-Inducible Factor 1α (HIF1α). These sources discuss HIF1α’s structure, regulation, accumulation, and overexpression in various cancers and metastases and its role in cancer metabolism, progression, and metastasis. Methods for studying or measuring tumor hypoxia are also included.	[10,14,23]
Angiogenesis and Tumor Vasculature	Includes sources focusing on the formation of new blood vessels (angiogenesis), key signaling molecules that stimulate this process, like Vascular Endothelial Growth Factor (VEGF), and the cells that form these vessels. The role of angiogenesis in tumor growth and metastasis is included, as well as methods for evaluating the tumor vasculature. The group also covers vascular patterns in specific tumors, like gliomas, and interactions between vascular cells and tumor cells.	[8,9,12,20,22,25,26,27]
Molecular Targets and Therapeutic Strategies	Includes sources discussing specific molecules or pathways that are explored as potential targets for cancer treatment and the different therapeutic approaches discussed in the sources. Examples include targeted inhibitors against Vascular Endothelial Growth Factor Receptor (VEGFR), Epidermal Growth Factor Receptor (EGFR), Platelet-Derived Growth Factor Receptor (PDGFR), cMet, MIF, Cyclooxygenase-2 (COX-2), and Monocarboxylate Transporters (MCTs). Therapeutic approaches mentioned include antiangiogenic agents (VEGF-Trap, Bevacizumab, FGF traps), chemotherapy (Temozolomide and Irinotecan), radiation therapy, and strategies to improve drug delivery.	[15,16,17,18,28,29]
Tumor Microenvironment and Cellular Interactions	Includes sources focusing on the complex ecosystem within and surrounding the tumor, involving various cell types and their interactions. It includes discussions on the tumor microenvironment, extracellular matrix (ECM), cancer stem cells (CSCs) and circulating tumor cells (CTCs), and the roles and interactions of different stromal cell types, such as endothelial cells, pericytes, astrocytes, microglia and macrophages, and glioma-associated vascular cells (GVCs). Specific interaction mechanisms, such as gap junctions, are also included.	[4,11,21,24,27,29,30]

## Data Availability

Not applicable.

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
