# Peer review of "Brain Metastasis: A Literary Review of the Possible Relationship Between Hypoxia and Angiogenesis in the Growth of Metastatic Brain Tumors"

_ijms, 2025, doi:10.3390/ijms26157541_

Round 1
Reviewer 1 Report
Comments and Suggestions for Authors
Comments to authors of manuscript (ID: ijms-3559008) by Colby L., et al. “Brain Metastasis: A Literary Review of the Possible Relationship Between Hypoxia and Angiogenesis in the Growth of Metastatic Brain Tumors”
Reviewer’s disclaimer: none of the suggested references are in any way related to me or my research or my institution.
While the topic is important, the review does not reach the level of analysis that is required for a good review. The paper is rather sloppy and superficial. The literature review is basically information one can get using different AI tools like elicit.com. Part 2 gives summary of each study authors included in this paper. In the literature reviews authors usually need to talk about main concepts and possible paradigm shifts referencing original research.
Conclusion about intratumoural heterogeneity is well known fact. There are more than two underlying molecular and cellular mechanisms driving the therapy resistance of brain metastasis and it would be good to mention at least major ones.
Major flaws:
Hypoxia alone is a big and wide concept. There are various types of hypoxias (acute, chronic, short etc.) (Lee, SCES., Pyo, AHA., Koritzinsky M. “Longitudinal dynamics of the tumor hypoxia response: From enzyme activity to biological phenotype” Science Advances, 22 Nov 2023, Vol 9, Issue 47; DOI: 10.1126/sciadv.adj6409).
In addition, VEGF also has different forms involved in angiogenesis. Besides VEGF there are other factors involved in angiogenesis in hypoxia like FGF and Ang-2.
The importance of localization of brain metastasis is overlooked – several papers mention different levels of hypoxia in different parts of brains (see Corroyer-Dulmont, A., Valable, S., Fantin, J. et al. Multimodal evaluation of hypoxia in brain metastases of lung cancer and interest of hypoxia image-guided radiotherapy. Sci Rep 11, 11239 (2021). https://doi.org/10.1038/s41598-021-90662-0).
Lots of old references where some newer ones should be used, as the information grows over the years (see detailed comments below).
Language must be improved, especially use of professional slang, e.g., “mets” as metastasis. MetS is also an abbreviation for metabolic syndrome. Literary review probably is literature review.
Comments:
Line 100-101: This paper considers the possibility of brain mets establishing a symbiotic relationship characterized by one area of the tumor intentionally maintaining a hypoxic environment to allow for continuous stimulation of angiogenesis via HIF1α and VEGF – it is really not a novel concept.
Line 35 – 37: “In this paper we provide a literary review of the relationship between hypoxia and angiogenesis in brain mets, exploring the possibility of this relationship with the help of commonly used therapeutic agents.” One really cannot understand what is meant by “possibility of this relationship with help of commonly used therapeutic agents”.
As for ref 1: Brain metastasis incidences... here is better reference (not related to me or my research): https://pmc.ncbi.nlm.nih.gov/articles/PMC11341633/ - J Neurooncol., 2024 Jun 19;169(3):457–467. doi: 10.1007/s11060-024-04748-6. “Incidence of brain metastasis according to patient race and primary cancer origin: a systematic review”.
Line 49-50. There is, in fact, a marked inefficiency to tumor metastasis, studies showing that, in order for metastasis to proceed, a specific set of steps must be undertaken, that if not followed in the correct order, will halt invasion and growth of the met (2) – What does “marked inefficiency of tumour metastasis” means? We probably have more detailed information nowadays for brain metastasis formation.
Peter Vaupel FR, and Paul Okunieff. Blood Flow, Oxygen and Nutrient Supply, and Metabolic Microenvironment of Human Tumors: A Review. Department of Radiation Medicine, Massachusetts General Hospital Cancer Center, Harvard Medical School, Boston, Massachusetts 02114. December 1, 1989;49, 6449-6465.
There are more recent reviews on formation brain metastasis. Also - Incomplete year of publication: 8. Isaiah J Fidler SY, Ruo-dan Zhang, Takahashi Fujimaki, Corazon D Bucana. The seed and soil hypothesis: vascularisation and brain metastases. The Lancet Oncology. 200 January;Volume 3(Issue 1):53-7
Citation for ref 10 should be without first names: Magar AG, Morya VK, Kwak MK, Oh JU, Noh KC. A Molecular Perspective on HIF-1α and Angiogenic Stimulator Networks and Their Role in Solid Tumors: An Update. Int J Mol Sci. 2024 Mar 14;25(6):3313. doi: 10.3390/ijms25063313. PMID: 38542288; PMCID: PMC10970012.
Line 79-80. This interaction allows for maintenance of a variety of different metabolic pathways, such as glycolysis and suppression of p53 to inhibit apoptosis (10). – HIF and hypoxia is so much more! See: Lee, SCES., Pyo, AHA., Koritzinsky M. “Longitudinal dynamics of the tumor hypoxia response: From enzyme activity to biological phenotype” Science Advances, 22 Nov 2023, Vol 9, Issue 47; DOI: 10.1126/sciadv.adj6409
Line 92: ...Topotecan and Irinotecan which inhibits Topoisomerase 1 disrupting replication as well as inhibiting HIF1α (17, 18),....
Ref 17 mentions Torotecan, however, it is a review, where there are references to actual research article about mode of action of topotecan (Topotecan, selective inhibition of topoisomerase I, disrupts the replication and transcription processes in the tumor cells, which leads to cell death. At the same time, it also inhibits HIF-1α by affecting RNA transcription and blocking the insulin-like growth factor-I (Beppu et al., 2005; Rapisarda et al., 2009).
Reference 18 is about irinotecan and has nothing to do with brain metastasis or HIF inhibition.
- Dervieux T, Meshkin B, Neri B. Pharmacogenetic testing: proofs of principle and pharmacoeconomic implications. Mutation Research/Fundamental and Molecular Mechanisms of Mutagenesis. 2005;573(1):180-94.
doi: https://doi.org/10.1016/j.mrfmmm.2004.07.025
Section 2. Line 112. “There were eleven studies that discussed the relationships of VEGF, hypoxia, and angiogenesis.”
Given the breadth of research across different disciplines, it's safe to say that hundreds to thousands of studies have explored these relationships. When doing search in ncbi.nlm.nih.gov using: VEGF AND hypoxia AND angiogenesis AND brain metastasis – one can get 196 hits. Of course not all will be relevant, however, I would argue there are probably more. Authors probably wanted to mention “there are 11 studies related to hypoxia, VEGF and brain metastasis, used in this review”.
Most of the literature review is just a summary of each paper chosen for this review.

Language must be improved, especially use of professional slang, e.g., “mets” as metastasis. MetS is also an abbreviation for metabolic syndrome. Literary review probably is literature review.
Reviewer 2 Report
Comments and Suggestions for Authors
Comments
The work of Colby L, et al. describes the influence of hypoxia and angiogenesis in the development of brain mets. The topic is interesting, and the methodology well designed. There are, however, some aspects that need to be addressed.
Major comments
- The introduction is not clear. First, state the problem. Second, describe the problem. Third, state the scope of the paper. Third, state the aim and objectives.
- Make the section more understandable and readable.
- Describe the search in the materials and Methods. Do not add just a figure.
- It is mandatory to add a table with the major findings of section 2.1, 2.2, 2.3. Add something, such as, relation, major finding, description, and reference.
Minor comments
Line 16 – Which transcription factors?
Line 42 – describe which chemotherapeutic agents
Line 51 – support this with a figure
Line 111 – This section is huge and only has one paragraph.
Line 170 – Figure 2 is not clear
Line 173 – Huge paragraph
Line 205 – Medications is not the right title for the section. Change for Chemotherapy, or Therapeutic protocols, or something similar. Huge paragraph.
Line 333 – Materials and methods is a figure?
Round 2
Reviewer 2 Report
Comments and Suggestions for Authors
Comments
The work developed by Lara Colby, et al. describes the importance of the hypoxia and angiogenesis in the growth of metastatic brain tumors. The work has improved significantly, however, there are some aspects that must be addressed before publication.
Major comments
- Justify the addition of Caroline Preskitt
Minor comments
Line 16 – add also the abbreviations, HIF1α and VEGF
Line 47 – What do you mean with “seeding of the blood vessels?
Line 49 – What is “cooption”?
Line 116 – The table need further improvements. A column with the key players, a second column with the explanation of the relation and then the references…
Line 194 – Too big… Which make it very difficult to understand
Line 225 – And about the European and American therapeutic guidelines? This section will benefit with a table with the major findings.
Round 3
Reviewer 2 Report
Comments and Suggestions for Authors
The authors replied succesffully to all my comments
Author Response
We thank the reviewer for their comments